

# Arbuscular mycorrhizal fungi increase crop yields by improving biomass under rainfed condition: a meta-analysis

Shanwei Wu[1,2,3], Zhaoyong Shi[1,2,3], Xianni Chen[1,2,3], Jiakai Gao[1] and Xugang Wang[1]

[1] College of Agriculture, Henan University of Science and Technology, Luoyang, Henan Province, China
[2] Henan Engineering Research Center of Human Settlements, Luoyang, Henan Province, China
[3] Luoyang Key Laboratory of Symbiotic Microorganism and Green Development, Luoyang, Henan Province, China

## ABSTRACT

**Background:** Rainfed agriculture plays key role in ensuring food security and maintain ecological balance. Especially in developing areas, most grain food are produced rainfed agricultural ecosystem. Therefore, the increase of crop yields in rainfed agricultural ecosystem becomes vital as well as ensuring global food security.

**Methods:** The potential roles of arbuscular mycorrhizal fungi (AMF) in improving crop yields under rainfed condition were explored based on 546 pairs of observations published from 1950 to 2021.

**Results:** AMF inoculation increased 23.0% crop yields based on 13 popular crops under rainfed condition. Not only was crop biomass of shoot and root increased 24.2% and 29.6% by AMF inocula, respectively but also seed number and pod/fruit number per plant were enhanced markedly. Further, the effect of AMF on crop yields depended on different crop groups. AMF improved more yield of N-fixing crops than non-N-fixing crops. The effect of AMF changed between grain and non-grain crops with the effect size of 0.216 and 0.352, respectively. AMF inoculation enhances stress resistance and photosynthesis of host crop in rainfed agriculture.

**Conclusion:** AMF increased crop yields by enhancing shoot biomass due to the improvement of plant nutrition, photosynthesis, and stress resistance in rainfed field. Our findings provide a new view for understanding the sustainable productivity in rainfed agroecosystem, which enriched the theory of AMF functional diversity. This study provided a theoretical and technical way for sustainable production under rainfed agriculture.

Corresponding author
Zhaoyong Shi,
9903105@haust.edu.cn

# INTRODUCTION

Rainfed agriculture is a farming type that relies on rainfall, which plays a dominant role in producing food for increasing world population (*Molle, 2008*). Rainfed areas cover worldwide 80% of the cultivated land, and contribute about 60% of crop production (*UNESCO, 2009*). The rainfed farmland get to more than 95% of the total cultivated area in

sub-Saharan Africa, 90% in Latin America, 75% in the Near East and North Africa, 65% in East Asia, and 60% in South Asia (*Giordano et al., 2012*), which ensures food security for mankind, especially in some developing countries (*Rosegrant et al., 2002*). With the more and more serious global warming (*Birara, Pandey & Mishra, 2018*; *Yadav et al., 2018*), the global food security also has been threatened by the impact of climate on crop productivity (*van der Linden & Goldberg, 2020*). *Ahmed, Fayyaz-Ul-Hassan & Zhang (2015)* confirmed that the climate change affected adversely crop yield in rainfed area. Rainfall and temperature are recognized as the two most important factors during climate changes (*Abera et al., 2018*; *Gebrechorkos, Hülsmann & Bernhofer, 2019*), which also influence the plant growth in rainfed cropland (*Gebrechorkos, Hülsmann & Bernhofer, 2019*). In subhumid and humid zones, rainfed agriculture generates high yields because of relatively reliable rainfall and inherently productive soils (*Molden et al., 2011*). However, arid and semiarid regions have experienced the low yield, which is a problem to be solved. *Licker et al. (2010)* estimated that winter wheat produced only 25% to 50% potential yields under non-irrigated condition comparing to irrigated field in global rainfed agriculture. *Jin et al. (2016)* also indicated that the wheat yield is 2.3 times higher in irrigated farmland than that in rainfed condition in Loess Plateau of China. On one hand, water stress may lead to stomata closure, which inhibits nutrient uptake (*Downton, Loveys & Grant, 1988*). On the other hand, water stress induces plant nutrient uptake and water use efficiency due to the decrease of microbial activity in soil (*Yi et al., 2007*; *Wang et al., 2017*). A lot of studies have testified that crop nutrition was also limited under rainfed condition in developing countries (*Salvagiotti et al., 2008*; *Setiyono et al., 2010*; *Qin et al., 2015*). Therefore, nutrient and water uptake were two vital factors for increasing crop yield in rainfed region (*Xu et al., 2021*). Besides, there are also several factors which lead to the reduction of crop productivity under rainfed condition, such as land degradation, nutrient depletion and biodiversity decrease.

It is important not only to increase the yield in rainfed area but also to protect soil biodiversity by taking sustainable management practices. Many management measures have been employed to enhance crop yield with the aim to ameliorate abiotic stress, such as soil mulching management (*Gan et al., 2013*), different tillage system (*Bakhshandeh et al., 2017*) and biological fertilizer application (*Karaca et al., 2013*; *Cavagnaro et al., 2015*). Among of them, arbuscular mycorrhizal fungal inoculation has been concentrated widely due to its functions in improving the water status of host plant in agroecosystem (*Bryla & Duniway, 1997*; *Askari et al., 2019*). *Hijri (2016)* clearly demonstrated the advantage of arbuscular mycorrhizal fungal inoculation on potato yield in large-scale production system. As a natural bio-fertilizer, Arbuscular mycorrhizal fungi (AMF) are paid special attraction owing to their important roles in improving nutrition of host plants and status of soil fertility (*Karaca et al., 2013*). AMF inoculation can improve plant growth through increasing nutrients absorption, photosynthesis (*Ruiz-Sánchez et al., 2010*) and water stress resistance (*Heidari & Karami, 2014*). Meanwhile, AMF inoculum is also an environment-friendly agronomic measure to enhance crop yield (*Celebi et al., 2010*), which is considered as a promising option in ensuring crop yield and food security in rainfed agriculture (*Rillig et al., 2016*; *Thirkell et al., 2017*). Numerous studies have

reported that AMF was able to improve the absorption of nutrients such as phosphorus, nitrogen, and zinc in plants (*Ardakani et al., 2009*). *Smith & Read (2008)* pointed out that AMF inoculation can supply up to 90% of plant P and 20% of plant N due to the hyphal networks in the soil formed symbiotic associations with host plant, which is also confirmed in Johnson's finding (2012) (*Johnson et al., 2012*). In particular, the impacts of abiotic stress such as drought, nutrient imbalance and temperature regimes on plant growth finally have decreased crop yield up to 70% (*Saxena et al., 2013*; *Kumar et al., 2020*). The resistances of plant which inoculated with AMF were enhanced by improving tissue hydration and stomatal conductance (*Augé, Toler & Saxton, 2015*) and photosynthesis (*Quiroga et al., 2017*; *Amirnia et al., 2019*) and alleviating oxidative stress (*Chitarra et al., 2016*; *Mirshad & Puthur, 2016*). Especially under water stress, AMF can improve water status of host plant and maintain osmotic balance (*Ruiz-Lozano, 2003*; *Porcel et al., 2006*; *Malfanova et al., 2011*; *Bárzana et al., 2014*; *Li et al., 2016*). *Wu & Xia (2006)* also drew a conclusion that AMF played a key role in improving crop yield in rainfed agricultural system by improving drought resistance of host plants. Additionally, the influence of biotic stress also leaded to yield losses such as bacterial, viral, nematode phytopathogens and herbivores (*Dowarah, Gill & Agarwala, 2021*). AMF protects host plants against different biotic stresses by acting alone or in synergy with other native microorganisms (*Dowarah, Gill & Agarwala, 2021*). Many researches also pointed out that AMF stimulates plant growth and yield through increase the tolerance to biotic stress (*Fiorilli et al., 2018*; *Bernaola & Stout, 2020*).

To evaluate the importance of AMF for crop yield under rainfed condition, we need a profound quantitative understanding. The mechanism that AMF can increase the crop yield under rainfed agriculture has been testified in some crops including wheat (*Zhu et al., 2017*), barley (*Espidkar et al., 2017*), soybean (*Suri & Choudhary, 2013*), and chickpea (*Erman et al., 2011*; *Sharma et al., 2019*; *Rezaie et al., 2020*). However, the quantitative estimation of predictor variables on crop responses to inoculation with AMF in rainfed agriculture is scarce on a global scale. Meanwhile, in our knowledge, only two quantitative synthesis has so far targeted the AMF effects on wheat (*Pellegrino et al., 2015*) and cereal crops (*Zhang et al., 2019a*) in field studies. For rainfed agriculture, there is no meta-analysis about AMF on crop yield and we are going to fill this knowledge gap. Therefore, we hypothesize that: (1) AMF inoculation can increase the crop yield in rainfed area; (2) AMF inoculation can enhance biomass in rainfed area. In this study, we verified these hypotheses based on global data.

## MATERIALS AND METHODS

### Establishment of database

This study followed the PRISMA guidelines (*Moher et al., 2016*) (Fig. 1). The published papers from 1950 to 2021 in the Web of Science™ (subscripted by Henan University of Science and Technology, Luoyang, Henan, China) have been searched. The Web of Science™ included multiple databases (Web of Science Core Collection, MEDLINE, SciELO Citation Index, KCl-Korean Journal Database and Russian Science Citation Index). The references cited in publications have also been retrieved. Two researchers
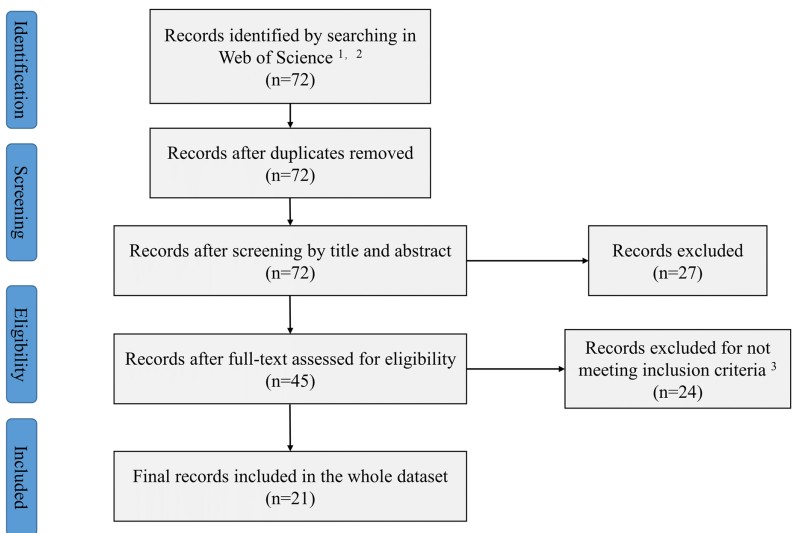

**Figure 1 Preferred reporting items for systematic reviews and meta-analysis (PRISMA) flow diagram.** [1]We used the terms rain*fed or non-irrigat* and mycorrhiz*. [2]Web of science include five databases (Web of Science Core Collection, MEDLINE, SciELO Citation Index, KCl-Korean Journal Database and Russian Science Citation Index). [3]We used the following inclusion criteria: (1) The articles had to be original research, (2) field studies under rainfed conditions, (3) involve an AMF treatment and a corresponding control and (4) contained replicated controlled trials.

(Shanwei Wu and Xianni Chen) independently performed the search strategy (Note S) following the methods in *Foo et al. (2021)*. All discrepancies were resolved through discussion with each other or consultation with a 3rd reviewer (Zhaoyong Shi). Our search terms were 'rain*fed OR non-irrigat* AND mycorrhiza*'. These searches resulted in 72 articles. To ensure representativeness and accuracy of the results, the following criteria were used to screen article for inclusion: (1) the articles had to be original research, (2) field studies under rainfed conditions, (3) involve an AMF treatment and a corresponding control and (4) contained replicated controlled trials. Based on the above criteria, 21 articles were selected. The information of yield, biomass, replications and other variables including plant, nutrition and physiological characteristics were extracted from the article we selected. The digitizing software (GetData Graph Digitizer v.2.20) were used for data extraction if the data were only available *via* graphs.

For a more detailed investigation, five groups of moderator variables related to yield were used as explanatory variables in meta-analyses following the methods in *Hoeksema et al. (2010)* and *Zhang et al. (2019a)*. We focus on the most studied crops in rainfed area including wheat, soybean, barley, chickpea, lentil, sorghum, yellow sweet clover, strawberry, tomato, olive trees, lavender, rosemary and thyme.

*Crop Species* had two levels with grain crops and non-grain crops. Chickpea (*Cicer arietinum* L.), lentil (*Lens culinaris* Medik), wheat (*Triticum aestivum* L.), barley (*Hordeum vulgare* L.), soybean (*Glycine max* L. Merrill) and sorghum (*Sorghum bicolor*) were grouped in grain crops. Yellow sweet clover (*Melilotus officinalis* L.), strawberry (*Fragariax ananassa*), tomato (*Solanum lycopersicum*), olive trees (*Canarium* spp.),

lavender (*Lavandula officinalis* L.), rosemary (*Rosmarinus officinalis* L.) and thyme (*Thymus vulgaris*) were grouped in non-grain groups.

*Crop Functional Group* had two levels: N-fixing crops and non-N-fixing. Chickpea, lentil, soybean and yellow sweet clover were grouped into N-fixing crops. Wheat, barley, sorghum, strawberry, tomato, olive tree, lavender, rosemary and thyme were grouped into non-N-fixing crops.

*Crop yield component* analysis is a general methodology aiming to probe in the yield-building factors. The effect size of seed number per plant, pod/fruit number per plant, seed number per spike, thousand seed weight and harvest index were calculated.

*Biomass* were grouped into shoot and root biomass.

*Plant nutrients* are not only the important indicators for plant growth and yield but also the best available approach to assess the mycorrhizal function. The resulting effect sizes of plant (shoot, leaf) nitrogen, phosphorus concentration (% of biomass) represented the AMF effect on nutrient status of plant tissues. The food qualities were indicated by the effect size of seed nitrogen, phosphorus concentration and uptake.

*Plant physiological characteristics* included proline and chlorophyll. Crop yield is dependent upon photosynthesis and the exchange of carbon metabolites from source to sink tissues (*Oiestad, Martin & Giroux, 2019*). As an important indicator of the growth and photosynthesis of plant (*Sun et al., 2021*), the leaf chlorophyll effect size were calculated. Additionally, the resistance of plant is an important indicator to plant growth under rainfed condition. Proline accumulation is responsible for plant resistance (*Sharma & Singh, 2016*), which used as a biochemical marker of abiotic stress in plants. Therefore, proline and chlorophyll content were contained as two vital indicators of plant stress tolerance and photosynthesis.

To deal with non-independence issue, four types of corrections were conducted as applied in previous publications: (1) For the effect of AMF on different plant species, one plant species during different years or different AMF species in the same article, the observations were considered to be independent (*Koide, 2003*), (2) For multiple studies from one author/lab, the observations were considered to be independent (*Koide, 2003*), (3) For the observations in the same article with different treatment, we use the two-way method followed by *Song et al. (2020)* to handle non-independence issue, (4) For studies presenting multiple observations in the same year, the observations were combined into one effect size value following a random-effects meta-analysis model (*Schütz et al., 2018*). Finally, we assembled the Global Dataset of Arbuscular Mycorrhizal Fungi and Crop Yields under Rainfed Agroecosystem in the Supplementary Data Sheets. The locations of studies from database were shown in Fig. 2.

## Calculation of effect sizes

The natural log response ratio (ln R) was used as effect size in our analyses to represent the AMF effect on yield. The effect size was calculated followed the following equation:

Ln R = ln $(X_i/X_n)$, with $X_i$ denoting the yield in an inoculated treatment and $X_n$ indicating the yield of the corresponding control. A positive ln R indicated a beneficial AMF effect on yield, while negative values represented a negative effect. The effect size of

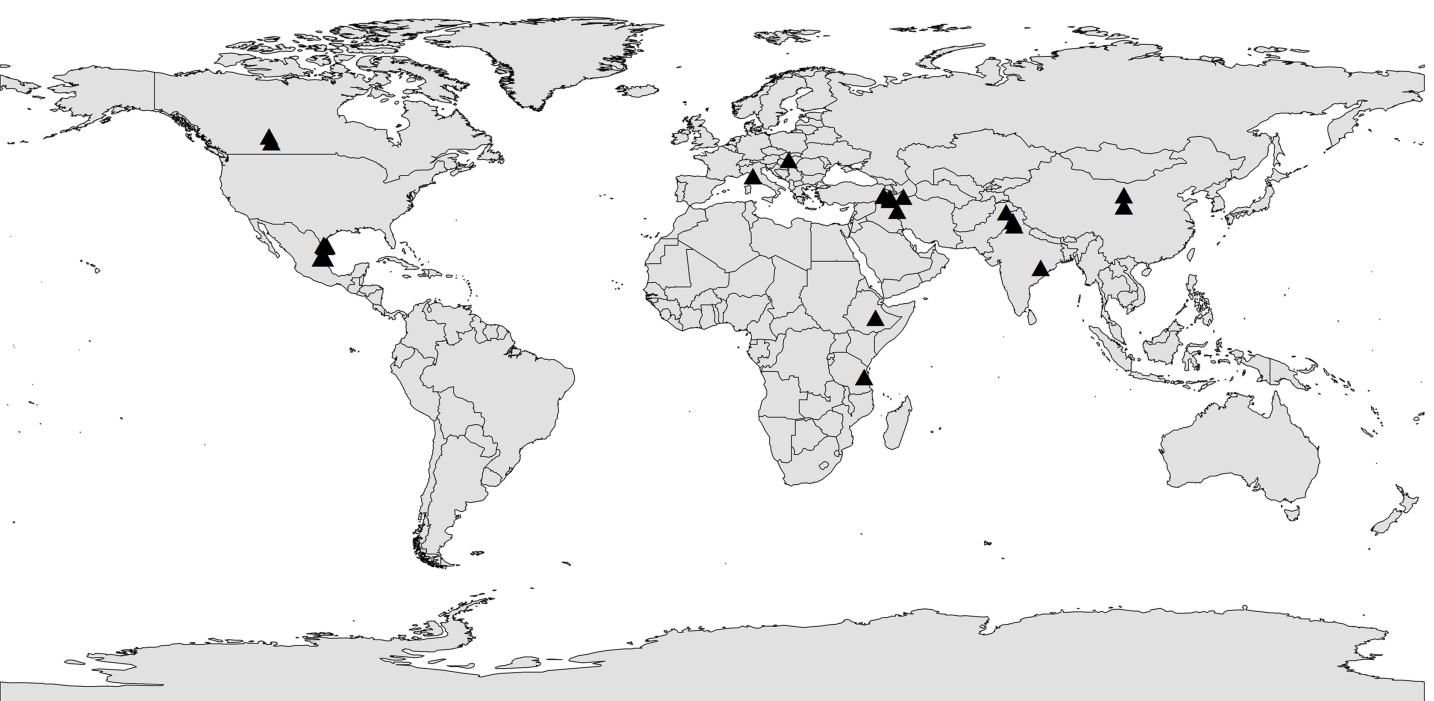

**Figure 2** **The locations of studies included in the global dataset of arbuscular mycorrhizal fungi and crop yields under rainfed agroecosystem.**
The black triangle represents the locations of studies.

AMF on yield component, biomass, nutrition, harvest index, chlorophyll, and proline were calculated in the same way.

The effect size of AMF was calculated by the overall weighted mean effect size according to the method described by *Hoeksema et al. (2010)* owing to the insufficient standard deviation (SD) or standard errors (SE). The weighted value of effect size was estimated according the method employed by *Hoeksema et al. (2010)*. The detail procedure including: (1) the reciprocal of the sum between AMF treatment replications and non-inoculated control replications (Rs) was calculated, (2) the maximum likelihood was estimated, (3) the weight of effect size was obtained by adding Rs and maximum likelihood. All the analyses were conducted in R v.3.4.1 (*R Core Team, 2017*) with 'METAFOR' package (*Viechtbauer, 2010*). The overall effect of AMF on crop yields and other variables was estimated with the rma.uni() function by a random effect model with a restricted maximum likelihood method.

To verify our analysis outcomes, sensitivity analysis was performed by using publication bias (Fig. S) (*Sterne & Egger, 2001*) and there were no patterns suggesting the existence of publication bias.

## Structural equation modeling

Structural equation modeling (SEM) is a statistical method to express the relationship between observation variables by using linear equation system. The advantage of SEM is estimate interdependence of several variables. SEM was used in this study to evaluate the relationships among AMF, biomass and yield. Simultaneously, the effects of nutrient,
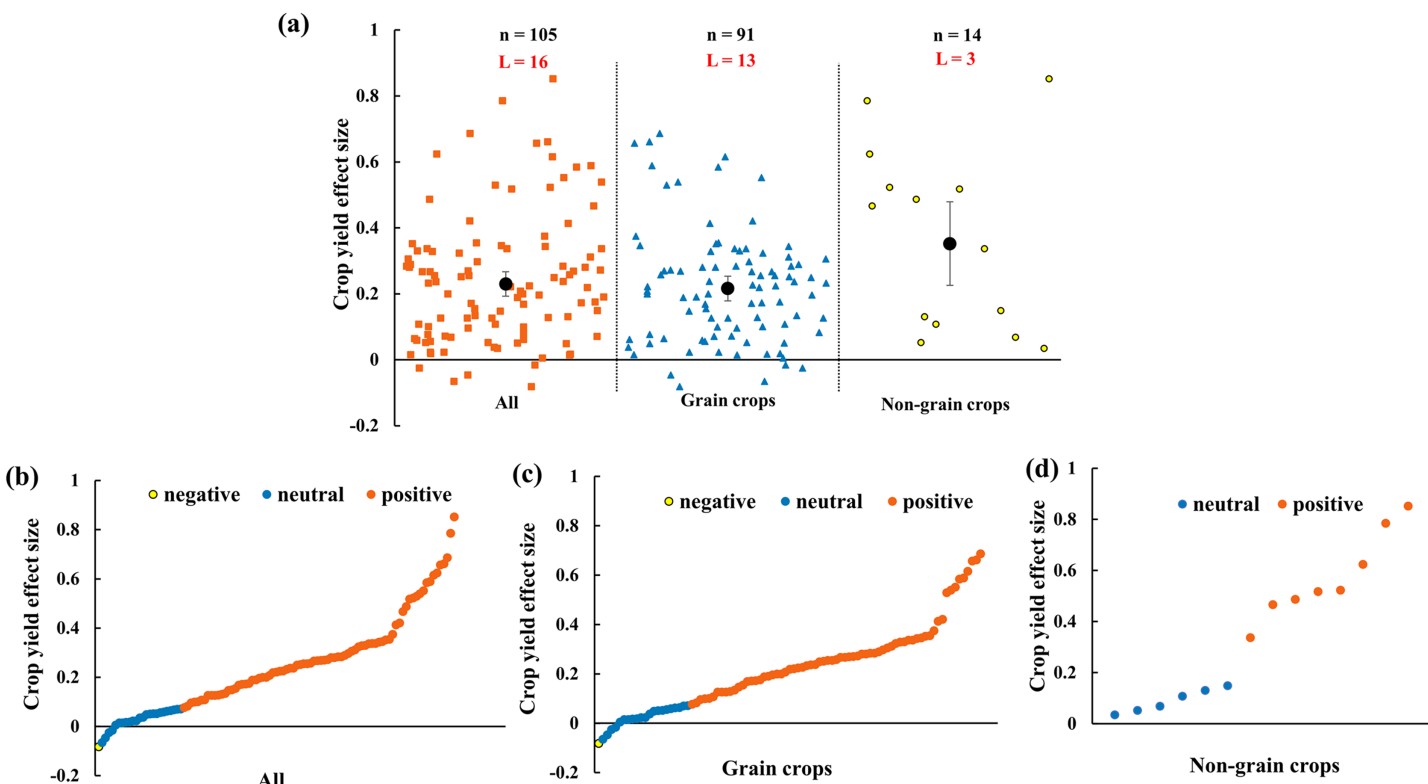

**Figure 3 Effect of arbuscular mycorrhizal fungi (AMF) on crop yields under rainfed condition.** (A) Overall effect on crop yield in all, grain crops and non-grain crops. Effects are displayed as weighted means and 95% Cis. 'n' represents trial numbers and 'L' represents literature numbers. The vermilion square, blue triangle and yellow point represent the original data distribution of all, grain crops and non-grain crops. (B)–(D) Effect size for each trial in (B) all (CI [16–30%]), (C) grain crops (CI [14–29%]) and (D) non-grain crops (CI [10–60%]). The vermilion, blue and yellow data point represent the positive, neutral and negative effect.

physiological characteristics, environmental resistance, crop yield component, biomass, and yield were analyzed. The standardized path coefficients (r) was calculated in R v.3.4.1 (R Core Team, 2017) with 'Iavaan' package according to the method described by *Jiang et al. (2019)* and *Rosseel (2012)*.

## RESULTS

### Effect of AMF on crop yields

The crop yield increases 23.0% (CI [16–30%]) by AMF inoculation in rainfed agriculture (Fig. 3A). To evaluate the function of AMF in different plant groups, grain crops and non-grain crops were differentiated. The results showed that both grain crops and non-grain crops yielded increase (Fig. 3A). For the distribution of every effect size of yield, the effect size changed from −8% to 85% with the 76.2% positive effect sizes (Fig. 3B). For grain crops, the positive, neutral and negative effect size respectively accounted for 75.8%, 23.1% and 1.1% (Fig. 3C). Due to the small sample size of non-grain crops, the effect size of yield only included positive (57.1%) and neutral (42.9%) (Fig. 3D).

When crops were classified into N-fixing crops and non-N-fixing crops, the results showed that both N-fixing and non-N-fixing crops yielded positive but N-fixing crops

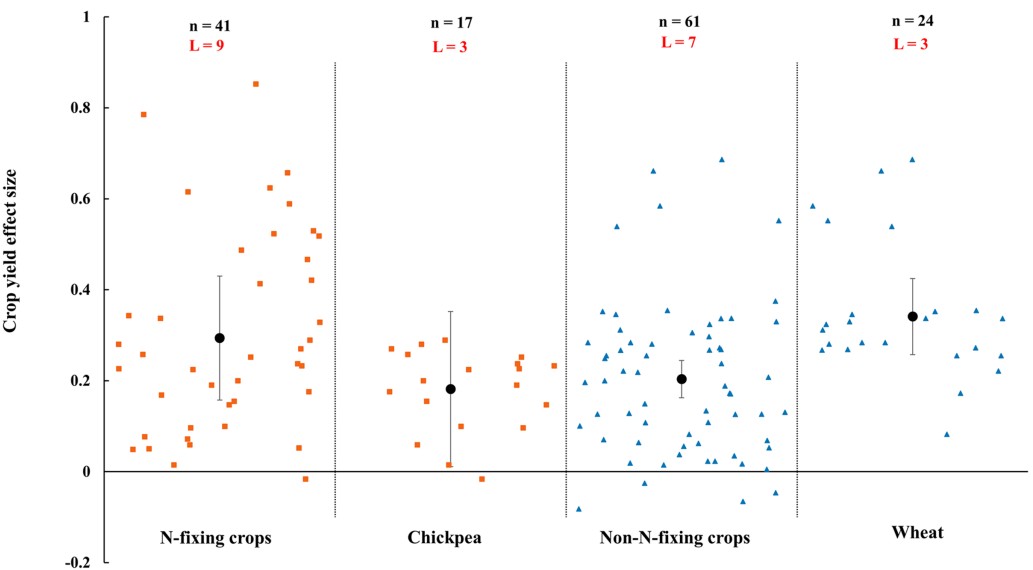

**Figure 4 Effect of arbuscular mycorrhizal fungi (AMF) on crop yields under rainfed condition for crop functional group (N-fixing crops and non-N-fixing crops).** Effects are displayed as weighted means and 95% Cis. 'n' represents trial numbers and 'L' represents literature numbers. The vermilion square and blue triangle represent the original data distribution of N-fixing crops and non-N-fixing crops.

(29.4%, increase, CI [16–43%]) was higher than non-N-fixing crops (20.4%, increase, CI [12–28%]) (Fig. 4). Among N-fixing crops and non-N-fixing crops, as the widely farmed and studied crops in rainfed area, chickpea and wheat were increased by 18.1% (CI [1–35%]) and 34.1% (Cl [18–51%]), respectively (Fig. 4).

The results in Fig. 5 showed that the AMF inoculation significantly increased the seed number per plant and pod/fruit number per plant under rainfed condition. But the effect size of seed number per spike and thousand seed weight were neutral. Compared with non-inoculated AMF, the seed number per plant and pod/fruit number per plant increased by 32.2% (CI [11–54%]) and 20.8% (CI [1–41%]) by inoculating with AMF.

## The effect size of AMF on biomass

The effect of AMF on biomass depended on the organs (shoot and root) and functional groups of host plants (Fig. 6A). Overall, the shoot and root biomass increased 24.2% (CI [15–33%]) and 29.6% (CI [16–43%]), respectively. The shoot biomass effect sizes of AMF were difference when crops were classified into grain and non-grain groups. The shoot biomass of non-grain crops increased with 54.9% (CI [33–77%]) while grain crops with the enhancement of 17.4% (CI [9–26%]). All the effect size in Figs. 6B and 6C showed that the positive effect size accounted for 73.8 percent of the whole shoot biomass effect and 81.3 percent of the root biomass.

When the plant functional groups were considered, both N-fixing crops and non-N-fixing crops exhibited positive response to shoot biomass by AMF inoculation under rainfed condition (Fig. 7). The shoot biomass of N-fixing crops was increased by 31.0% (CI [18–44%]) which is higher than non-N-fixing crops (16.9%, increase,

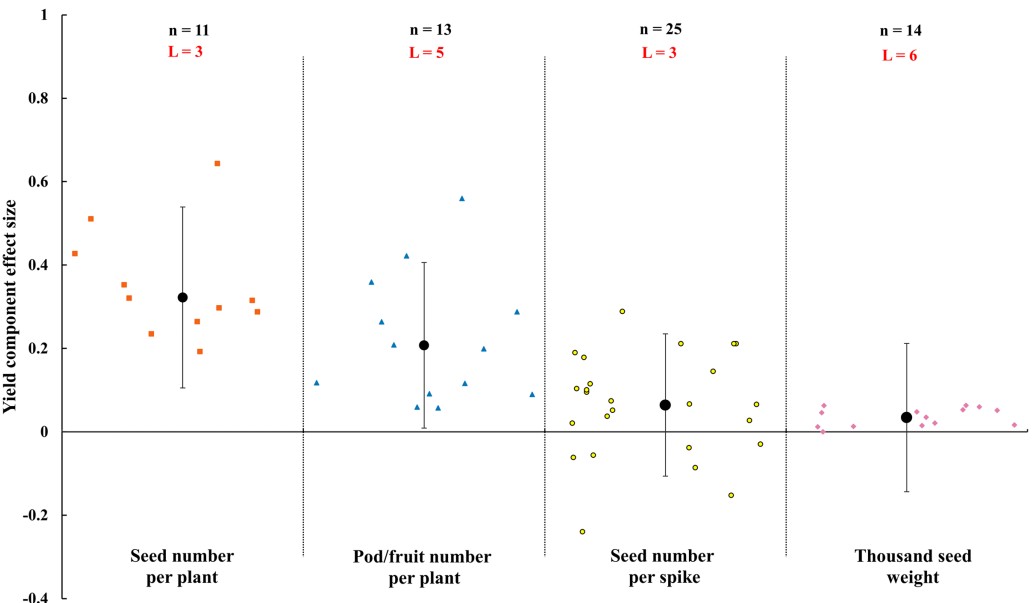

**Figure 5 Impact of arbuscular mycorrhizal fungi (AMF) on crop yield component under rainfed condition (seed number per plant, pod/fruit number per plant, seed number per spike and thousand seed weight).** Effects are displayed as weighted means and 95% Cis. 'n' represents trial numbers and 'L' represents literature numbers. The vermilion square, blue triangle, yellow point and reddish purple diamond represent the original data distribution of seed number per plant, pod/fruit number per plant, seed number per spike and thousand seed weight.

CI [5–28%]). Among N-fixing group, shoot biomass of dominated plant species yellow sweet clover and chickpea were increased 69.9% (CI [46–95%]) and 20.8% (CI [7–35%]) by AMF in this study (Fig. 7). There was also different between N-fixing crops and non-N-fixing crops in root biomass (Fig. 7). The root biomass effect size of N-fixing crops was neutral while the non-N-crops was positive (29.6%, increase, CI [15–44%]).

## The effect size of AMF on physiological status

AMF inoculation apparently improved physiological status of host plant (Figs. 8–10). In rainfed agriculture, the proline content was decreased more than 67.1% (CI [−92% to 43%]) by inoculated AMF. However, AMF inoculation increased 40.6% (CI [11–70%]) in chlorophyll content of host plant under rainfed condition (Fig. 8).

The N and P in shoot were increased significantly by AMF (Fig. 9) when the N and P concentration in shoot and leaf were estimated. The effect of AMF on shoot and leaf nutrition presented difference with the positive and neutral, respectively. The shoot P concentration increased by 46.0% CI [26%–66%], which is higher than shoot N concentration (31.9%, increase, CI [9–55%]). The neutral effect of AMF was observed in concentration of leaf P and N with 12.7% (CI [−11% to 37%]) and 15.4% (CI [−11% to 42%]). Moreover, there was a large different between nutrition concentrations of shoot and seed by AMF inoculation. The effect size of seed P and N uptake and concentration were neutral (Fig. 10).

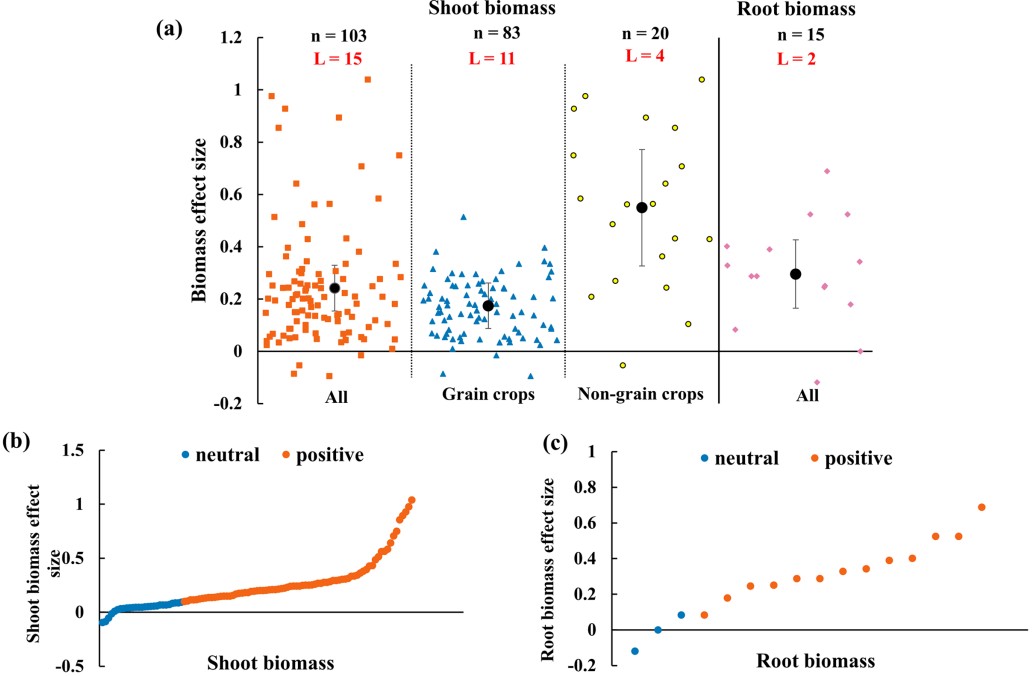

**Figure 6 Effect of arbuscular mycorrhizal fungi (AMF) on shoot and root biomass under rainfed condition.** (A) Overall effect on biomass in all, grain crops and non-grain crops shoot biomass and all root biomass. Effects are displayed as weighted means and 95% Cis. 'n' represents trial numbers and 'L' represents literature numbers. The vermilion square, blue triangle and yellow point represent the original data distribution of all, grain crops and non-grain crops in shoot biomass. The reddish purple diamond represent the original data distribution of root biomass. (B & C) Effect size for each trial in (B) shoot biomass (CI [15–33%]), (C) root biomass (CI [16–43%]). The vermilion and blue data point represent the positive and neutral effect.

## Relationships among AMF, physiological and yield

Structural equation model analysis indicated that AMF had a positive effect on yield *via* its neutral effect on harvest index and positive effect on pod/fruit number, whereas AMF had positive effect on biomass *via* its influence on nutrition, chlorophyll, and abiotic resistance of the host plant. The direct positive effect of nutrition on biomass in response to AMF inoculation(r = 0.18) was weaker than the direct positive effect of chlorophyll on biomass(r = 0.99), whereas the direct effect of proline on biomass was negative (r = −0.98).

Biomass had a direct positive effect on yield (r = 0.72), while the direct positive effect of pod/fruit number and harvest index on yield was not significant. The results also showed that the direct positive effect of biomass on pod/fruit number and harvest index was not significant (Fig.11).

## DISCUSSION

In this study, we explored firstly the AMF effect on crop yields in rainfed agricultural ecosystem. Our results provide insight on how AMF effects on crop yield under rainfed condition and inform management practices in rainfed agriculture. However, there are limitations in this study. Some literatures may be missed because the word of rainfed was employed although studies were carried out under rainfed condition. The results showed

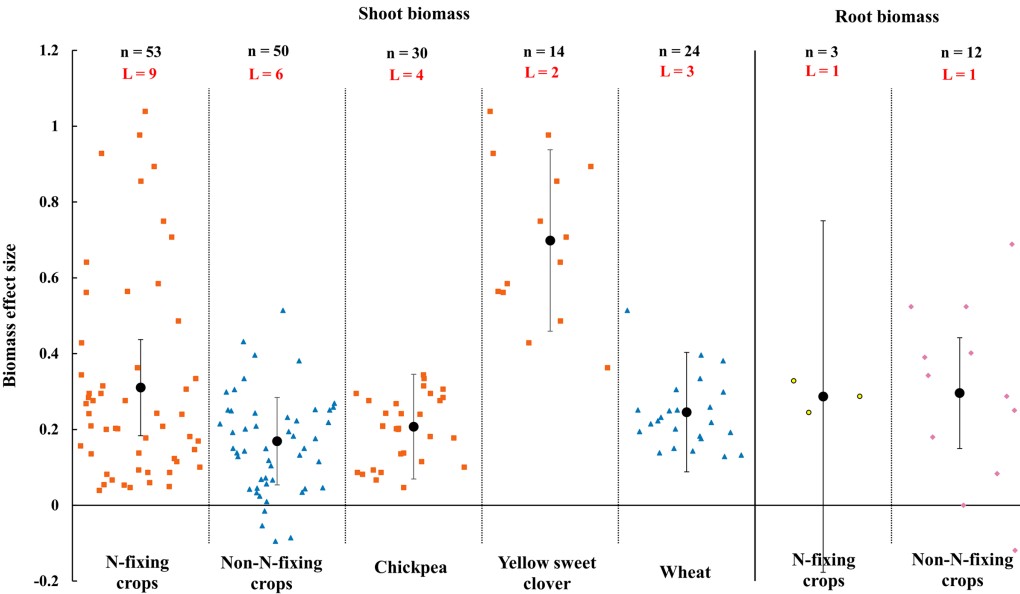

**Figure 7 Effect of arbuscular mycorrhizal fungi (AMF) on shoot and root biomass between N-fixing and non-N-fixing crops under rainfed condition.** Effects are displayed as weighted means and 95% Cis. 'n' represents trial numbers and 'L' represents literature numbers. The vermilion square and blue triangle represent the shoot biomass original data distribution of N-fixing crops and non-N-fixing crops. The yellow point and reddish purple diamond represent the root biomass original data distribution of N-fixing crops and non-N-fixing crops.                         

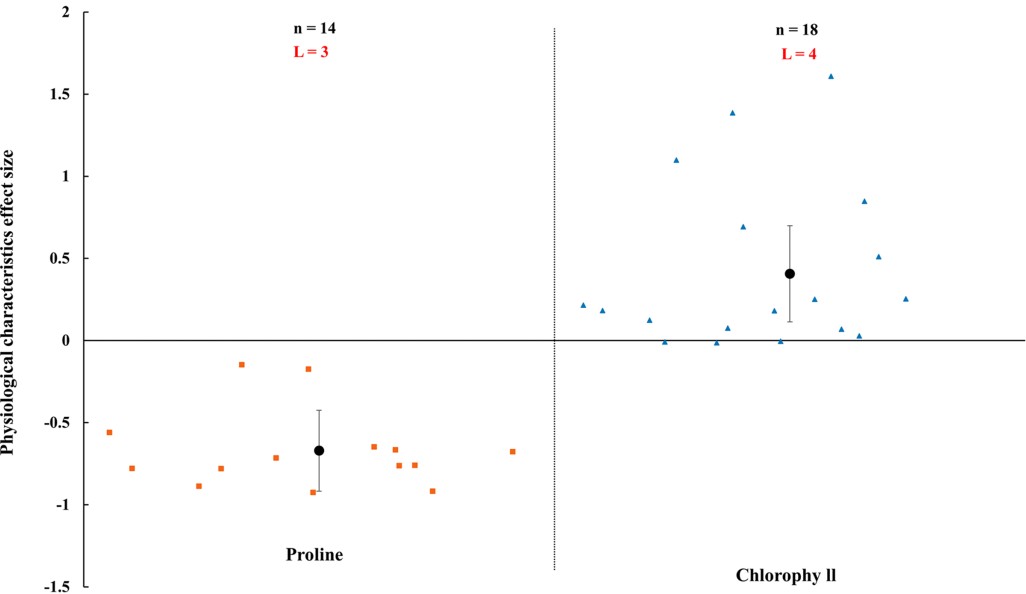

**Figure 8 Effect of arbuscular mycorrhizal fungi (AMF) on plant proline and chlorophyll under rainfed condition.** Effects are displayed as weighted means and 95% Cis. 'n' represents trial numbers and 'L' represents literature numbers. The vermilion square and blue triangle represent the original data distribution of proline and chlorophyll.                         

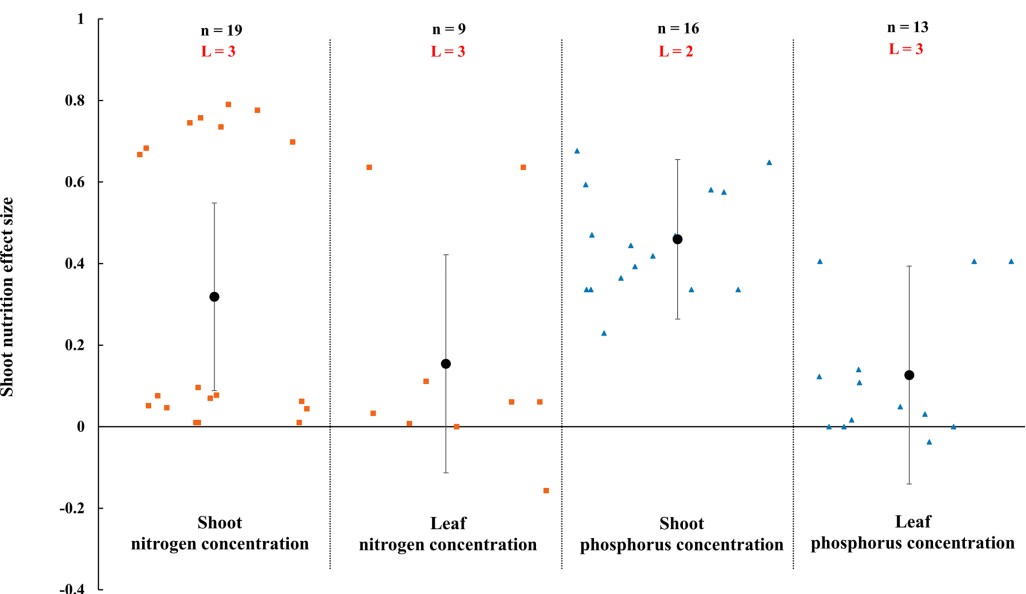

**Figure 9  Effect of arbuscular mycorrhizal fungi (AMF) on nutrient concentration of shoot and leaf under rainfed condition.** Effects are displayed as weighted means and 95% Cis. 'n' represents trial numbers and 'L' represents literature numbers. The vermilion square and blue triangle represent the original data distribution of nitrogen and phosphorus concentration of shoot and leaf.

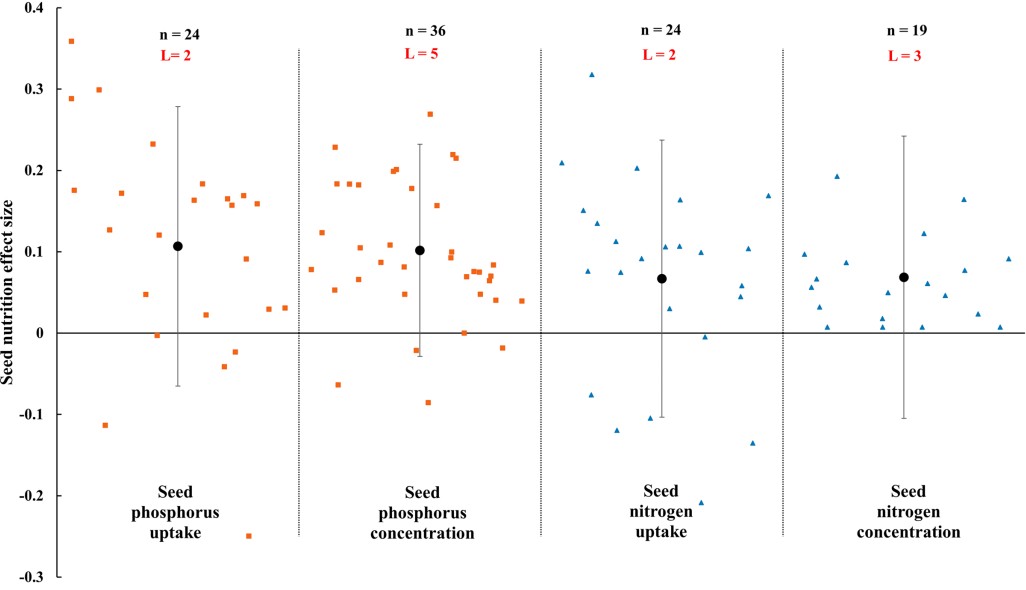

**Figure 10  Effect of arbuscular mycorrhizal fungal (AMF) on nutrient concentration and uptake in seed under rainfed condition.** Effects are displayed as weighted means and 95% Cis. 'n' represents trial numbers and 'L' represents literature numbers. The vermilion square represents the original data distribution of phosphorus concentration and uptake of seed, the blue triangle represents the original data distribution of nitrogen concentration and uptake of seed.

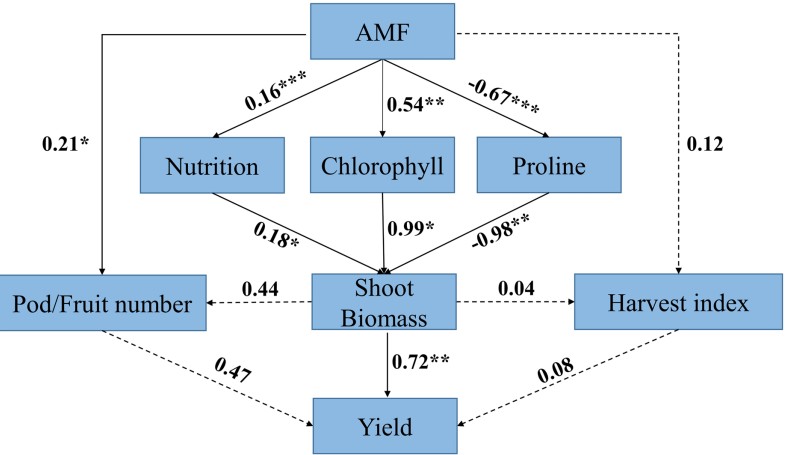

**Figure 11 Structural equation model (SEM) analysis of relationships among AMF, response ratio of physiological characteristics (*i.e.*, nutrition, chlorophyll and proline), biomass, yield component (*i.e.*, pod/fruit number and harvest index) and yield.** Black solid arrows represent significant positive or negative effects. Black dashed arrows represent nonsignificant paths. Number near lines represent standardized path coefficients (r), *** means $P < 0.001$; ** means $P < 0.01$; * means $P < 0.05$, respectively.

that AMF increased 23.0% crop yield under rainfed condition (Fig. 3A), which is in correspondence with previous numerous studies (*Erman et al., 2011*; *Zhu et al., 2017*; *Saadat, Pirzad & Jalilian, 2019*). AMF symbiosis with plants exerted a variety of positive roles for crop yield in rainfed agricultural system (*Al-Karaki & Al-Raddad, 1997*; *Bryla & Duniway, 1997*; *Al-Karaki, McMichael & Zak, 2004*; *Wu & Xia, 2006*). On one hand, AMF inoculation improved plants roots ability to uptake more water from the soil, as well as improved plants root hydraulic properties through higher flexibility of mycorrhizal roots to switch water transport pathways under water stress (*Fiorilli et al., 2018*; *Bernaola & Stout, 2020*). On the other hand, the host plant resistance to drought and plant growth were improvement by AMF symbiosis (*Cho et al., 2006*; *Li et al., 2019*). Similarly, *Zhang et al. (2019a)* also concluded that AMF inoculation in field led to 16% increase on cereal yield, which is the same tendency to our study. Numerous studies have confirmed that the dependency of host plant on AMF was higher in stress environments than in no-stress environments (*Latef et al., 2016*; *Hu et al., 2017*; *Mohammadzadeh & Pirzad, 2021*). Furthermore, There are differences between grain crops and non-grain crops by AMF, which probably because the positive effect of AMF on yield depends on the plant species (*Tarraf et al., 2015*). In addition, the effect of AMF inoculation on crop yield was diverse in different functional plant groups. AMF improved more yield of the effect size of N-fixing crops by AMF was higher than non-N-fixing crops (Fig. 4). For N-fixing crops, AMF symbiosis promote rhizobia accumulation in the rhizosphere of host plant and eventually result in increasing yield and biomass (*Wang et al., 2021*). AMF increase the seed number per plant and pod/fruit number per plant under rainfed condition (Fig. 5). The reason may be attributed to the synergistic effect of AMF has increased the number of pods by increasing the absorption of water and nutrients (*Rezaie et al., 2020*). Although numerous of researches reported that AMF

improved crop yield (*Espidkar et al., 2017*; *Zhu et al., 2017*; *Sharma et al., 2019*), some studies involved that the yield benefits of AMF in agroecosystems are often overstated (*Duan et al., 2010*; *Ryan & Graham, 2018*). In conventional agriculture, AMF did not play a vital role due to high fertilizer input and tillage treatments (*Ryan & Graham, 2002*). Because the AMF community structure was greatly impacted by the management measures (*Jansa et al., 2002*, *2003*). However, AMF was beneficial to soil and crop in an organic agroecosystem.

Numerous studies have testified that AMF increased crop biomass accumulations (*Pellegrino et al., 2015*; *Shao et al., 2018*). Our results supported previous finding, which showed that AMF increased 24.2% shoot biomass and 29.6% root biomass under rainfed condition (Fig. 6). This probably caused by the following reasons. Firstly, AMF promotes plant growth due to improving water status and the availability of soil nutrients by the extension of mycorrhizal hyphae under rainfed condition (*Hazzoumi et al., 2015*; *Püschel et al., 2016*). Secondly, AMF mycelium enhanced the uptake scope of roots to nutrients and water (*Ruiz-Lozano, 2003*; *Zhang et al., 2019b*). In addition, AMF increased host plants biomass (*Gosling et al., 2006*; *Jia-Dong et al., 2019*) because the root of host plant absorbed more water that lead to leaves stomata remaining open longer under rainfed condition (*Zhu et al., 2012*). Further, shoot biomass effect size of N-fixing crops by AMF was higher than non-N-fixing crops (Fig. 7), which is possibly lead by stimulating nodulation and nitrogen fixation owing to AMF symbiosis in legume crops (*Abbott & Robson, 1977*). *Powell (1976)* also confirmed that the growth and phosphate uptake of legume plant completely dependent on AMF infection in phosphorus deficient soil. There were differences in root biomass between N-fixing and non-N-fixing crops (Fig. 7). AMF increased root biomass of non-N-fixing crops, which also confirmed by *Pellegrino & Bedini (2014)*. It maybe depends on not only the inactivation of nutrient uptake pathway *via* root hairs and epidermis but also functional diversity of AMF when non-N-fixing inoculated with AMF (*Klironomos, 2003*; *Smith, Smith & Jakobsen, 2004*). However, AMF inoculation had no effect on root biomass of N-fixing crops, which may be caused by root nodulation (*Suri & Choudhary, 2013*).

A decrease in proline and an increase in chlorophyll were observed when inoculated with AMF under rainfed condition (Fig. 8). Proline as usual osmoprotectants which stabilize cellular membranes and sustain turgor pressure (*Umezawa et al., 2006*). The accumulation of proline were increased when plants were under environmental stress (*Farhad et al., 2011*). The plant inoculated with AMF had lower proline content that presented a negative effect size in Fig. 8, compared with non-inoculated plants. It is confirmed that AMF plants have stronger tolerance under rainfed condition. Numerous studies have shown AMF protect their host plant from various environmental stresses such as drought and metal pollution (*Kavi Kishor et al., 2005*; *Wu & Xia, 2006*; *Shi et al., 2018*). The results related to a protection mechanism against abiotic stress by AMF plants. There may be other strategies to deal with environmental stress in AMF plants, such as upregulating the antioxidant defense system and synthesis of osmolytes (*Al-Arjani, Hashem & Abd_Allah, 2020*), which had been also confirmed by numerous previous studies (*Porcel & Ruiz-Lozano, 2004*). *Rahimzadeh & Pirzad (2017)* reported that the

chlorophyll content was enhanced in mycorrhizal plants, which is correspondence with our study (Fig. 8). The increase of chlorophyll in AMF inoculated plant may be associated with the mobilization of the ions (*Amirnia et al., 2019*; *Ludwig-Müller, 2000*).

The effects of AMF on plants nutrients content have been widely reported due to its importance in rainfed agriculture. There is a significant increase in N and P concentration of shoot in Fig. 9. Higher nutrient uptake in mycorrhizal plants might be attributed to the wider absorption surface provided by the hyphae and improvement of soil hydrolytic conductivity, which lead to absorb more nutrients in host plant under stress condition (*Smith et al., 2010*; *Estrada-Campuzano, Slafer & Miralles, 2012*; *Rahimzadeh & Pirzad, 2017*). For example, The mycorrhizal plants can use more soluble phosphate than non-inoculated plants by improving nutrient status from rock phosphate (*Gyaneshwar et al., 2002*). Besides, AMF also can acquire N and transfer it to host plant by decomposing organic and inorganic material (*Govindarajulu et al., 2005*; *Pérez-Tienda et al., 2012*). AMF inoculation presented no effect on leaf nutrition according to our research (Fig. 9). There is also a little study pointed out that AMF infection in the field was apparent ineffectiveness to plant nutrition uptake (*McGonigle & Fitter, 1988*). There is evidence that the stimulation of AMF plant growth is suppressed in non-sterile soil by fungivorous microarthropods grazing the external mycelium (*McGonigle & Fitter, 1988*). There were differences in nutrition effect size between seed and shoot. AMF had a positive effect on shoot nutrition (Fig. 9) and a neutral effect on seed nutrition (Fig. 10), which is similar to *Erman et al. (2011)*. Compared with non-inoculated plant, AMF enhanced nutrition concentration by improving availability of nutrition (*Rokhzadi & Toashih, 2011*; *Habibzadeh et al., 2013*).

Structural equation model analysis showed that the relationship between AMF, biomass and yield, which indicated that AMF increased yield by increasing biomass under rainfed condition (Fig. 11). AMF synergistic interaction increased plant growth by providing the essential nutrients for host plant (*Lingua et al., 2013*). For example, AMF inoculation increased N, P and K uptake by plants under water deficit conditions (*Malfanova et al., 2011*). Especially under drought conditions, AMF inoculated improved plant growth and water status due to the higher stomatal conductance in host plants than the control plants (*Naseri et al., 2013*). This possibly related to the hyphal extensions of AMF that allow higher hydraulic conductivity than non-AMF (*Askari et al., 2019*). Therefore, symbiotic relationship between AMF and host plants played a beneficial role in improving yield of the host plant under rainfed condition.

## CONCLUSIONS

AMF obviously increased crop yields under rainfed condition. The effect of AMF on crop yields depended on crop functional groups in rainfed agroecosystem. Our study highlighted that AMF increase crop yield by improving shoot biomass in rainfed agriculture. The shoot biomass of inoculated plant enhanced by improving plant nutrients, photosynthesis and stress resistance. Our findings provided a new view for understanding the sustainable productivity in rainfed agroecosystem, which enriched the theory of AMF functional diversity.

### Funding

This work was supported by the National Natural Science Foundation of China (No. 32171620 and No. 31670499), the Program for Science & Technology Innovation Talents in Universities of Henan Province (18HASTIT013), the Scientific and Technological Research Projects in Henan province (192102110128), the Key Laboratory of Mountain Surface Processes and Ecological Regulation, CAS (20160618) and the Innovation Team Foundation (2015TTD002) of Henan University of Science & Technology. The funders had no role in study design, data collection and analysis, decision to publish, or preparation of the manuscript.

### Grant Disclosures

The following grant information was disclosed by the authors:
National Natural Science Foundation of China: 32171620, 31670499.
Program for Science & Technology Innovation Talents in Universities of Henan Province: 18HASTIT013.
Scientific and Technological Research Projects in Henan province: 192102110128.
Key Laboratory of Mountain Surface Processes and Ecological Regulation: CAS 20160618.
Innovation Team Foundation: 2015TTD002.
Henan University of Science & Technology.

### Competing Interests

The authors declare that they have no competing interests.

### Author Contributions

- Shanwei Wu conceived and designed the experiments, performed the experiments, analyzed the data, prepared figures and/or tables, authored or reviewed drafts of the paper, performed the search strategy, and approved the final draft.
- Zhaoyong Shi conceived and designed the experiments, performed the experiments, analyzed the data, prepared figures and/or tables, authored or reviewed drafts of the paper, referee, and approved the final draft.
- Xianni Chen performed the experiments, analyzed the data, authored or reviewed drafts of the paper, performed the search strategy, and approved the final draft.
- Jiakai Gao analyzed the data, prepared figures and/or tables, authored or reviewed drafts of the paper, and approved the final draft.
- Xugang Wang performed the experiments, analyzed the data, prepared figures and/or tables, authored or reviewed drafts of the paper, and approved the final draft.

### Data Availability

The raw data is available in the Supplemental Files.

## Supplemental Information

Supplemental information for this article can be found online at http://dx.doi.org/10.7717/peerj.12861#supplemental-information.

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
