# Peer review of "Arbuscular mycorrhizal fungi increase crop yields by improving biomass under rainfed condition: a meta-analysis"

_PeerJ, doi:10.7717/peerj.12861_

## Round 0.1 · original submission · Major Revisions

You have four experts who have read your manuscript, they have presented several constructive suggestions for your work.

Reviewer 1 ·

Basic reporting

no comment

Experimental design

The hypothesis and addressed research gap should be clarified more precisely. Please find my comments for the Introduction section under “4. Additional comments”.

I identified some issues with the meta-analytical approach (e.g. imprecise reporting of data set building, not state of the art statistics, lack of addressing non-independence issue among effect sizes and lack of application of sensitivity analysis). Please find my comments for the methods section under “4. Additional comments”.
I have no expertise on structural equation modeling.

Validity of the findings

The data table comprising the effect size values was provided but the information about the included articles is missing. Please provide the reference list and indicate which article is associated to which data row.

The discussion has substance but is weak at some points. Please find my comments for the Discussion section under “4. Additional comments”. The conclusion section is just one sentence.

Additional comments

Title:

It is not clear from the title that this is a meta-analysis. Please add this term. You could also add it to the keywords. By this, you article can be easier found by a WoS search.


Introduction:

Line 67: I wonder how important AMF really are in agriculture. I guess they have their role in organic/ low input agriculture, but under conventional agricultural practices with high fertilizer input and tillage AMF play very likely only a minor part in improving crop production.

Line 75: What do you mean by “all kind of biotic and abiotic” constraints? This statement is very unspecific. It sounds like AMF can solve all plant related issues. That is not true. In the second half of this sentence you gave specific examples of what the AMF have proven to be capable off. This is a more precise and clear way to praise their beneficial symbiotic functions.

Line 86: This is again unspecific. Please make clear what exactly you mean and want to focus at in this work. AMF have many functions. You probably want to focus on functions related to crop productivity and nutritional quality?
What do you mean by "effect size of AMF"? The effect size represents the magnitude of a treatment effect compared to an associated control effect. Please rephrase and clarify.

Line 88: So there is no meta-analysis on AMF-mediated effects on crop production/growth etc. for agricultural systems under rainfed irrigation only? If this is true, then just state it here clearly. This is your identified research gap which you want to fill with your work. So make it clear so that the readers understand what is your new knowledge gained from this work you contribute to the science community.

Line 91: You do not mention in the methods that you also collected data on non-rainfed systems. How are these data incorporated in the following analyses? Were they just used in a sub-analysis or were they included in all analysis? Please clarify this very important point.


Methods:

Line 95. Excellent. Reporting guidelines are vital.

Line 96: Did you consider alternative terms to identify articles about this topic? Is there really only the term "rain*fed" used to describe water management without artificial irrigation? Especially in the context of applied science which affects humans (here in the form of food production), extra care has to be taken to make the meta-analysis as comprehensive as possible. What was your search term building strategy? Please have a look into this article which demonstrates with worked examples how to develop search strings following best practices. (Foo et al. 2021 https://doi.org/10.1111/2041-210X.13654)

Also, state more clearly what settings you chose for your search. You give search date and publication year range, but you do not say if you run a topic or title search, which database in WoS you used (WoS is the platform, not the database. CoreCollection is a database in WoS. So which did you use? Please specify.), if you preferred/ excluded any languages or citation indexes. Subscriptions with WoS vary widely across different research institutes and hence the available publication years, databases and citation indexes, could you please add your institution/university name in this section. Then it is transparent with which subscription setting you did your search.

Line 97: This is already a very small number of articles to begin with which makes the following meta-analysis quite challenging. Did you consider searching other possible sources for articles e.g. other databases or the reference lists of identified matching articles or even gray literature? Each article is important to increase statistical power and to correctly estimate AMF effects.
Also, please state how many articles were in the end matching your inclusion criteria. I know you give this information the PRISMA flow diagram, but also give this information here.

Line 98: I highly appreciate your effort to dual screen and cross-check during the screening process. This is done very rarely but a very important routine. Excellent.

Line 105: Your hypothesis focus only on biomass and yield, then why did you also collect data on proline and chlorophyll? And what do you mean by “nutrition”? Please clarify.

Line 116: You explain which effect size metric you chose and explain how to interpret it. But you do not mention the effect size variance at all. How did you calculate it? How did you deal with missing variance data? How did you deal with missing information on replication in experiments? (Which are needed to calculate the effect size variance.) By the way, how did you extract data from e.g. figures and graphs? Please add this important information to this section.

Line 119: Please add more information about the data you used for your effect sizes: e.g. biomass (What biomass data did you included, shoot, root, total?). Give concise information to allow readers not familiar with harvest index, proline and Co. to understand why these parameters are interesting and what the data you included in your data set looked like. How did you decide that different aspects of a metric can be combined (a hypothetical example: how to decide to include root and shoot biomass data in the biomass effect size but exclude total biomass data)?

Also, add information about your moderators (explanatory variable): e.g. N- and non-N-fixing crops. You mention these in your results section but never introduced them in the methods. The same for grain crops and non-grain crops in Fig. 2. Give concise rationales why these moderators or subsets are meaningful and interesting for your project and what is included in each group (e.g. which crop species are included in the grain and non-grain crops groups).

Line 121: How did you weighted the effect size, with the effect size variance data, I guess? Please add a statement.

Line 124: I recommend using rma.mv() because this allows inclusion of the data hierarchy and addressing the non-independence issue (see below). This is a mixed-effects meta-analysis model. In the random factor you can include the information of articles comprising specific effect sizes means (rows in your data table). Please, find this article on the topic by Song et al. 2020 https://doi.org/10.1002/ecy.3184. They explain very well the possible options to address this problem of non-independency of effect sizes and why it is so important to take care off it.

Ok, following your statement in line 103 “491 pairs of observations” where extracted from 17 articles. So, each article gave you multiple data pairs. This is absolutely normal in ecological MAs but you need to account for the resulting non-independence issue in your dataset or else your data analyses are flawed.

The non-independence issue. From your methods it is not clear to me if you have corrected for non-independence issues in your dataset. There are two typical sources: 1) multiple data from one study (which is definitely the case here) and 2) multiple studies from one author/lab (which is quite likely the case here). As mentioned above you can apply rma.mv()-models to account for hierarchical structure in your dataset, but it is still useful to “reduce” your dataset as far as possible. With this I mean, only have the absolute minimum of effect sizes (rows in the data table) per study. E.g. When potentially matching data for multiple harvests are presented you can decide to only use data from the last harvest. (You just have to report your decisions and give a rationale.) When data are given for different AM fungal species (actually I am surprised that you did not mention AMF genera/species as moderator variable) you could “merge” these data by applying a “within-article meta-analysis model”. You could even apply a phylogenetic model (Anderson 2015 https://doi.org/10.1111/nph.13693). Please check out this paper explaining this important issue (Song et al 2020 https://doi.org/10.1002/ecy.3184).
Concerning the dependency of articles published by the same author/lab, I recommend to you the paper by Moulin & Amaral (2020; doi: 10.1002/jrsm.1430). Correcting for this source of “bias” will improve the robustness of your model outcomes.
These methods to target the non-independence issue are extremely important especially for meta-analysis on small numbers of articles. You artificially increase your statistical power by including all ~ 500 single effect size values without properly merging them as far as possible and/or accounting for data structure via a mixed-effect MA model (rma.mv).

Line 124: Please, add the package name "metafor" and its citation.

I apologize if I missed it, but you seem not to apply any sensitivity analyses to test for biases or the robustness in general of your data. A very popular tool is the publication bias. With this you can evaluate how likely it is that you have missed a publication (either it was never published or you really physically missed it) and weather you over-or underestimate potentially the true effect with your data collection. Another option are the leave-one-out-approaches. Here, you test the individual impact of data points or whole studies, by excluding them from the analysis, re-run the analysis, include the data points or whole studies again, delete the next one, and so on and so on. By this you can identify data points or studies with disproportional impact on your results. There are many other techniques, but at least one should find its way into your manuscript to further improve its trustworthiness and robustness.
When using rma.mv() many diagnostic function from the metafor package are not applicable anymore. However, there is the Eggers Regression Test for publication bias test which can be applied as a simple sensitivity analysis.


Results:

Line 135: The increased crop yield for non-grain crops should be treated with caution. As you state some lines later, this effect bases on a limited number articles and trials. So, you should carefully phrase your conclusions based on data with such low statistical power.

Line 136: You plotted every single effect size. You already calculated them or else you could not run any analysis.

Line 152: I get confused here. Did you collect only data for wheat and chickpea? No. But it reads like it. You really need to make clear in the methods what plant species you included in your data base. And you should better add also a section in the methods where you explain the analyses you are doing in this project. In the results section the reader should have an idea what to expect but here new subset groups/moderators and subset analyses keep popping up. This makes it hard to follow you.
So you chose wheat and chickpea in this subset analysis because they are the most abundant plant species in your dataset? Please give a rational for your decision. The same applies later to the line 190.

You only focus on the AMF-mediated effect on crop growth/productivity/nutrition but neglect the performance of the AMF completely. E.g. do you have data on AM fungal root colonization? This could be used as a potential proxy of the intensity of the plant-host symbiosis and could explain some patterns you find. More colonization higher AM fungal mediated productivity effects? Crop species specific differences in percent AM fungal root colonization?


Discussion:

Line 246: The global scale part is not convincing when presenting data comprised from 17 articles. Better delete that term.

Line 251: “Glomalin” or glomalin-related soil proteins“ are a rather critical group of substances. As discussed in this article (Rosier CL, Hoye AT, Rillig MC. 2006. Glomalin-related soil protein: assessment of current detection and quantification tools. Soil Biology & Biochemistry 38: 2205-2211.) detection methods for glomalin are not as specific as had previously been assumed. Such I do not see that this group of compounds should be discussed here.

Line 253: Could you extract data on hyphal length. By this you could test this in your very own data set and check if it really can explain the beneficial impact of AMF in the rainfed systems.

Line 255: This comparison (whose effect size is bigger) is a bit odd. Your data base is way more limited than that of this Zhang meta-analysis, hence pointing on magnitude effect size percentages is not valid. In general, you need to be careful with overemphasizing your findings. Your meta-analysis is interesting and important but it also bases on “only” 17 articles and hence give us just a first glimpse on an area of application for AMF in rainfed agricultural systems but it does not give us percent values which are set in stone.

Line 256: Sorry, this argument is also odd. Yes, you include more crops but by this you also increase heterogeneity in your data set. It is difficult to compare the performance of AMF symbiosis with members of the Poaceae and Fabaceae. Poaceae crops can have very vast and fine root systems reducing the importance of the AM fungal extraradical hyphal network for nutrient uptake, while Fabaceae crops can form a tri-partite symbiosis with AMF and Rhizobia giving them the option under nutrient limited conditions to overcome P and N limitations. It is not an advantage or disadvantage of a synthesis to a have narrower or wider focus. So for me your argument is invalid.

Line 290: Yes, this information would be nice to have in the methods section when you explain why you include which plant parameters as effect sizes in your analyses.


Figures

The figures look a bit blurry. This is a resolution issue, I guess. Please upload your nice figures with a higher resolution. I like your clear figure layout and color scheme. Very aesthetic.
Please add also the number of articles contributing to each group.
Add the effect size to the y-axis. In each graph it is written “Effect size” but which one. This should be indicated clearly and should not require reading the figure legend. Please add something like e.g. response ratio of biomass or biomass effect size… Be creative.

Fig. 3: Add the 95% confidence intervals to your effect size means. Or how did you decide which effect size is negative, neutral or positive? Now I’m a bit confused. I do this by checking if the 95% CI is overlapping with the zero line (line of no effect). This applies to all the effect size range plots.
Fig. 6: Why did you abandon here your color scheme and colored everything in red. Stay consistent.

Could you combine some figures? E.g Fig. 3 and Fig. 2 would fit nicely together showing the distribution of negative to positive single effect sizes and overall model outcomes.


Figure legends.

Why did you present SE and not the 95% confidence intervals which are given by the metafor model outcomes?


Data table.

Please provide information on the articles you used to build your data table. Also indicate which article is associated to which data row.

Reviewer 2 ·

Basic reporting

Abstract
- Underscore the scientific value-added to your paper in your abstract. Your abstract should clearly state the essence of the problem you are addressing, what you did and what you found and recommend. That will help a prospective reader of the abstract to decide if they wish to read the entire article.
- Lines 27-28: This sentence is not clear.
- Line 31: The sentence ‘Shoot biomass had a direct positive effect …’ should be deleted.
Introduction
- Line 43: Wikipedia is not a reliable source for reference in the article.
- Please add more information about climate change and increasing world temperature and drought area.
- the linkage between introduction paragraphs is missed.
- Lines 46-47: The sentence should be deleted.
- Lines 53-54: Please add another reasons for decreasing nutrient uptake under drought stress conditions; for example, low mobility, decreasing micro-organisms activity and etc.
- Please subject the manuscript to review made by English Native speaker.
- What is the innovation of this study? Please add in text. Many studies previously reported the effectiveness of AMF on the plant performance and productivity especially in drought stress conditions.
Materials and methods
- Lines 95-96: The authors used the paper that included two factors of AMF and Rain-fed or used simple mentioned factors effects?
- Line 130: Please add the method of standardized path coefficients calculating.
- The analysis method is not complete.
- The papers were used in this study was belonged to only one plant?
Results
- Please check again the increasing or decreasing percentage rate.
- Line 227: What is the SEM?
- Why the effect of the AMF in different climates (such as different between arid and semi-arid region) has not been studied.
Discussion
- In this section, the authors firstly reported the obtained results and after that noted the similar results of previous studies. The main reasons for increasing and decreasing of measured traits is missed. Actually, I don’t see any discussion in this section. This section should be deeply revised.
- What is the novelty of this study? The previously studies professionally reported the positive effects of AMF inoculation in plant performance. The novelty of this study is missed.

Experimental design

All comments are reported in the first section.

Validity of the findings

no comment.

·

Basic reporting

The manuscript deals with broadly researched and quite a recurrent topic, but it is done in a transparent and easily understandable fashion. Quality of writing and presentation is generally high (with just a few syntax/vocabulary items to be improved). However, I was not able to locate a list of the specific studies included in the meta-analysis (including literature references), which should definitely be done - for example by including references to each ana every row in the raw data file (Excel table in the supplement).

Experimental design

This contribution is a meta-analysis aiming at providing a global picture on AM fungal inoculation effects in rainfed agriculture. Whereas this is a worthwhile and valuable and broadly interesting question/objective, this has previously been approached many times and there also are other reviews/meta-analyses available. I am afraid the approach taken (using "rain*fed" as a keyword) is much too restrictive and the search returns only a minutious fraction of the relevant science that has already been conducted and reported. This is because a large part of the world such as Europe and North America produces lots of food under rainfed agriculture - but does not call it so. My own search using keywords "arbuscul* AND mycorrh* AND field and inoc*" returned 1621 research articles (compared to 55 articles recovered by authors of the current manuscript using their search), and these should be all thoroughly screened to recover more or less the complete story, including such important crops like maize and potatoes, for example, which are currently not at all represented in the manuscript. Besides, meta-analyses like McGonigle (1988) collated results of 78 field inoculation studies and Hijri et al (2016) reported results of 231 field trials just with potatoes should be a great guide, but these are not even cited in the current manuscript.

Validity of the findings

Whereas the results are in line with previous observations and quite well presented, I consider this only a preliminary efforts, too early to be considered for publication in an international journal. This is because a large majority of previously published and relevant research has been missed by too restrictive/not quite relevant keyword search, effectively missing many previous studies from Europe and North America. Europe and North America is also missing from the list of regions presented in lines 41-43, where they should definitely be included (because lots of the research has previously been generated there and they are still large regions of the Earth).
To provide a balanced view, more critical literature on employing AM fungal inoculation (e.g., Ryan et al 2018) should also be included in the discussion, next to the overly positive opinions by Rillig, Thirkell and others. It should also be clearly stated that every soil (or nearly every soil) has an active indigenous AM fungal communities already present and inoculation effectively brings "invaders" to the soil. The inoculation is thus effectively always representing a competitive interaction between native and incoming genotypes/communities. There is hardly any soil on the planet devoid of AM fungi (unless previously sterilized/disinfected, which is really not a widespread practice as yet).
Specific comments:
line 52: there are many regions in the world (particularly in middle/northern Europe) where water is not limiting yields, and rain provides sufficient (or even excessive) water. So the global validity of the research from Mediterranean climate zones should be critically evaluated and the existence of regions where water is sufficient should be clearly acknowledged.
line 251: currently there is an intensive debate about what a glomalin or GRSP actually is and who produces it (several opinion papers recently published in high IF journals). The writing should be adapted to be less authoritative/definitive in this regards.
line 253: there is really very thin or no experimental evidence that AM fungi would transport water (in spite of widely spread myths). There are some good papers showing that drought tolerance of mycorrhizal plants is often higher than tolerance of nonmycorrhizal plants, though. However, this is probably not relevant to the fields where all mycotrophic plants (inoculated or not) are mycorrhizal, due to indigenous AM fungal communities (see above). Thus the writing should be very cautious/specific at this point.

Reviewer 4 ·

Basic reporting

The paper presents a meta-analysis of data collected in the literature in order to demonstrate that symbiotic associations with Arbuscular Mycorrhizal Fungi (AMF) are beneficial to crop productivity with particular emphasis on crop biomass production and grain yield in a variety of crops. In addition, the aim of the study was to show that such a beneficial effect is species-specific, notably if legumes and non-legumes are considered. Additional information is also provided on the physiological impact of the symbiosis on representative physiological traits related to Phosphorus (P) and nitrogen (N) metabolism.
The study is moderately original, since a significant number of studies presenting such kind of analysis have been published over the last decade, including impact on yield and mineral nutrition. The novelty of the present study should be more clearly emphasized with respect the studies cited by the authors and to the published literature that was not cited in the paper.
The introduction starts and is largely focused on rainfed agriculture which is not really the main aim of the study. The introduction should be primarily focussed on the importance of AMF with respect to agricultural practices and crop productivity.
In the Introduction a number of references are a bit old (e.g. 2008-2009); there are many more recent references on rainfed agriculture and on the importance of AMF on crop productivity.

Experimental design

One of the major problems in the presentation of the paper is that, although a database is presented as Supplementary Data, one would like to see in this database references to the papers used for the meta-analysis (1 to 133 e.g. in a different sheet). In the supplementary Data set, the units for the different measured traits must be indicated. In line, with this comment Figure 1 provides very limited information and is not really necessary for the understanding of the paper, just mention it in the text.
One would like also to see in the various figure legends a brief description of the crops used to perform the analysis together with a way to have access to the corresponding data in the dataset.

Validity of the findings

- Unless I missed something, I do not see clearly the information we gain from the results presented in Figure 2 compared to that of Figure 1. Please explain a bit more in the Results section.
- Line 209: what do you mean by uptake by seeds? An uptake of nutrient is performed by the roots and is generally expressed at the whole plant level.
- In the Discussion there are a number of speculations related to water absorption when plants are colonized by AMF. However, nothing is presented on this aspect in the Results section.
- I am not sure that the interpretation of the lower of Proline in AMF inoculated plants. If they are more resistant to a water stress, higher amounts of Proline could be expected. Please clarify by citing and discussing more extensively the literature in the field.

Additional comments

Minor comments:
- Line 27, explain what is a crop functional group.
- Line 28 and throughout the text: the term effect size is not very clear to me and thus not the most appropriate. See also line 36 and in the Supplementary Data set.
- In the abstract, one would like to have a minimum of information on the crops used to performed the analysis.
- Line 54: what means N resorption?
- Line 57: management measures is not clear, please explain.
- Line 78: It is drought or water stress, not arid stress.
- Line 88: what is a normal grain production system?
- Line 116: indicate which type of yield.
- Line 227: Write Structural equation analysis at the beginning of the sentence.
- In the legend of figure 2: grain crops and not grin crops.
- In the Supplementary Data set it is Olive trees and not Olive tress.

---

## Round 0.2 · Minor Revisions

The reviewer ranged from rejection to accept. Please read the reviews carefully and respond accordingly.

Reviewer 1 ·

Basic reporting

yes

Experimental design

I have some minor questions concerning the methods. Please find my comments under "4. Additional comments"

Validity of the findings

yes

Additional comments

I thank the authors for the thorough revision. I just have a few comments left, please find these below.

I recognized that the English language of your text is a bit rough. As English is also not my native language, I would like to recommend to you to either ask an English native speaker or a language editing services to improve your text.

Introduction:
L 114: Maybe better write that no meta-analysis on that topic is available so far and that you going to fill this knowledge gap.
L 116: Could you maybe add a map with the study locations? In R there are several functions that allow you to plot decimal degree data for longitude and latitude on a world map. Most paper give location/GPS data for their field sites and in the worst case you could use the affiliation location of the first author. Then you could also highlight, for which rain-fed regions in the world no data could be found for your meta-analysis. These “data white spots” are a very important information when it comes to priorities future research (funds).

Methods:
As this point is not clear for me after reading the methods, I would like to ask if you only included field studies. I imagine that rain-fed systems are “outside” in the field but you never made a statement about it. Or did I overlook it.
If it really is only field studies, you should state it because the biological relevance and generalizability of your analysis outcomes would drastically increase. The majority of the crops you comprised in your analysis are cultured in outdoor agricultural systems and not under controlled greenhouse conditions.

L 173: I’m not convinced by the use of the fixed-effects meta-analysis model. I agree that merging the data from the same year into one effect size, is the right step to reduce the non-independence issue. But by using the fixed effects model you assume that the data for these different observations share the same true effect. This is unlikely, because these observations differ in some way from each other and because of that they were presented as individual data. Imagine in year 1 biomass data is given for sandy soil and clayey soil. These two observations have inherent heterogeneity because they differ in texture (and everything else that gets affected by texture). So, you have to account for this variability and this is done by random-effects models even on the level of the same article. This will give you higher values for your effect size variance, but this approach is more robust.

L 175: Please add here where this data table can be found (e.g. supplementary data of this article…).

L 203: Please add the publication bias outcomes to your supplementary data, thus people can see that there is no case of publication bias evident. You show the plot in your response on my comments from the last round.

Results:
L 299: There is a point missing: “12.7% (CI=-011-0.37)” should be “12.7% (CI=-0.11-0.37)”

Discussion:
L363-365: I cannot follow your thoughts here. First you write that AMF effects on yield in (conventional) agriculture is often overstated. I agree on that point. In especially conventional agriculture AMF play almost no role, due to high fertilizer input and soil disturbance (e.g. tillage). These management practices tend to reduce community richness and functionality leaving only “pioneer-type” AMF species in the soils. On the other hand, in an organic system, with well aggregated soil and active soil biota community, AMF can indeed provide valuable service to soil and crops.
So your second sentence about tillage, seems a bit detached from the prior sentence. Maybe you could rephrase these liens to make your statement clearer.


Others:
I like your protocol on the search string development (in your response to reviewer comments). Maybe you want to consider to add it to your supplementaries. People rarely show how they came up with their search terms. This adds a new layer of transparency and reproducibility to your work.

·

Basic reporting

The article is well written and (formally) well presented. Comments made previously by a number of reviewers have been addressed, though the critical weaknesses of the meta-analysis (search keywords) remain unchanged, resulting in the dataset being thin and not covering the entire range of available literature.

Experimental design

Search keywords did not cover the field sufficiently. Whereas authors only identified a handful of papers quoting "rainfed", my previous ciriticism was that there is lot more literature which actually address rainfed agriculture in the broadest definition of the term, which were not covered because the authors of those original papers did not use the word "rainfed". The authors of the presented meta-analysis now clearly write that (in their definition) rainfed agriculture occupies 80% of worlds cropland (beginning of the abstract) - but these 80% of the worlds croplands have not been representatively covered in the meta-analysis. The focus of the study is thus not clear, although the authors give, in their response, some other arguments such as in their understanding the rainfed agriculture is "naturally drought-prone croplans". Such definition is not given in the article, though, nor it is generally understandable/acceptable. So I dearly miss the specific focus of this meta-analysis, and the delimitation from other rainfed croplands which have not been covered only because they did not write "rainfed" in their texts. This is not sufficiently justified. Hundreds of inoculation trials from non-irrigated agriculture have been published and these ought to be covered shall the ambition of the presented paper be "representative". Besides, the claims presented in the text that synthesis of results from rainfed conditions has not been made so far is incorrect for reasons that I brought up previously and now again.

Validity of the findings

Given the thin dataset which is not corresponding to available knowledge, the validity remains quite limited - although the conclusions are in line with expectations. Unless the few papers which were included have a common denominator AND could clearly be distinguished from the other inoculation trials done under rainfed conditions elsewhere, but not calling it so, I still consider this manuscript a preliminary effort to what its aims declare and what should in fact be done. Just an idea - maybe include all literature on AMF inoculation under field conditions, and filter out those that were done under "irrigated" conditions - most of the remainder will correspond to what we all understand under "rainfed" conditions. I am sure there will be many more papers remaining in the pot.

Reviewer 4 ·

Basic reporting

No comment

Experimental design

No comment

Validity of the findings

No comment

Additional comments

I am satisfied with both the answers to my comments and the relevant modifications to the manuscript which has been greatly improved.

---

## Round 0.3 · accepted · Accept

You have met most of the suggestions of the reviewers.